# SUPERVISED OPTIMIZATION OF A CONTEXT-AWARE COLOR TRANSFER MODEL

## ABSTRACT

The task of image color style transfer aims to apply the color characteristics of a reference image to a content image while preserving its texture and structural integrity. However, two key challenges hinder effective training: (1) the scarcity of high-quality ground truth (GT) images for supervised learning, and (2) color distortions from suboptimal feature fusion. To address the first issue, we propose a novel GT generation strategy—the first systematic method, to our knowledge, for producing high-quality GT images. A source image is recolored into two variants, and identical regions are randomly cropped to form a content image, a style image, and a structurally aligned GT image with distinct color styles, enabling reliable and precise supervision. For second issue, we propose the Context-Aware Color Transfer Network (CANet). Previous methods, due to the absence of GT images, often focused on more precise color space mapping to improve color fidelity while overlooking the role of network architecture. In contrast, we are the first, to our knowledge, to introduce a spatial-channel attention mechanism into the task of color style transfer. Specifically, CANet processes the content and style images through separate downsampling extractor to extract texture and color features, which are then fused via a spatial-channel attention module for more accurate and consistent transfer. An image reconstruction module further reintegrates texture, reducing degradation and preserving structural integrity. By combining these two innovations, our method significantly outperforms state-of-the-art approaches, as extensive experiments demonstrate clear advantages in both quantitative evaluation and visual quality.

## 1 INTRODUCTION

In photographic workflows, environmental factors such as harsh sunlight and haze often degrade image quality, making color adjustment essential. Traditional manual editing is labor-intensive and time-consuming, while tools like filters Sun et al. (2017); Yim et al. (2020) and LUTs Zeng et al. (2020); Liu et al. (2024) still require careful tuning to ensure coherence. Image-to-image color style transfer provides a promising alternative by adapting the color distribution of a reference style image to a content image, while preserving structural and semantic integrity. This greatly reduces the burden of manual adjustment in post-processing workflows.

Similar to general style transfer tasks Chen et al. (2021); Chung et al. (2024); Zheng et al. (2024); Li (2024), the scarcity of paired ground-truth (GT) data for image color transfer remains a fundamental bottleneck. Constructing such data typically requires manual recoloring under strict structural alignment, which is both costly and difficult to scale. As a result, many existing methods rely on complex loss function designs or sophisticated training strategies, limiting their ability to simultaneously preserve structural content and semantic consistency.

Due to the lack of GT training pairs, early handcrafted-statistics methods Reinhard et al. (2001); Xiao & Ma (2006) are computationally efficient but lack semantic understanding, often producing flat or unnatural results. More recent convolution-based approaches Li et al. (2018); Yoo et al. (2019); Ho & Zhou (2021); Chiu & Gurari (2022); Ke et al. (2023); Fang et al. (2024) improve stylization quality but still rely mainly on global color transformations, failing to ensure semantic-level region consistency (e.g., sky-to-sky, foliage-to-foliage). Some recent methods Larchenko et al.

(2025); Li et al. (2025) emphasize mapping the color space of style images onto content images, but they overlook the importance of network architecture and still rely on convolution as the backbone.

As illustrated in Figure 1, to enable supervised training, we propose a novel GT generation strategy based on the observation that local regions within a single image often share a consistent color style. Specifically, a source image is recolored into two stylistic variants, from which identical spatial regions are extracted as the content and GT images. In addition, a random crop from one variant is used as the reference style image. This approach enables the automatic construction of reliable, structurally aligned GT pairs without manual annotation, while preserving real-world color coherence.

Considering that recent color transfer methods Li et al. (2025); Larchenko et al. (2025) mainly focus on more precise color space mappings but still rely on convolution-based backbones, while attention mechanisms Liu et al. (2021); Yao et al. (2019); Wang et al. (2023a); Fu et al. (2024) have already been widely adopted in other computer vision domains, we are the first to employ attention as the backbone in this field. Specifically, we propose CANet, which adopts a dual-branch design: content and style images are independently processed through progressive downsampling

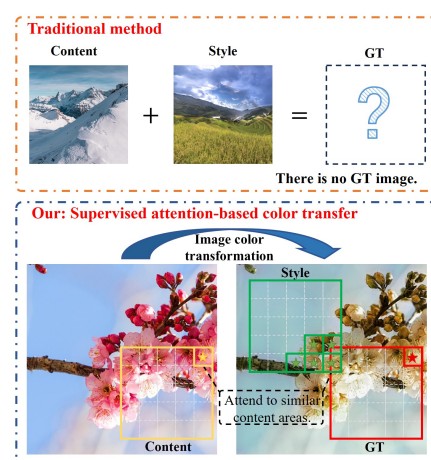

Figure 1: An example of the content image, reference style image, and the generated ground-truth (GT) image using our automatic strategy. The content and GT regions are spatially aligned.

(×16) to extract texture and color features. These features are then fused via a spatial-channel attention module to achieve context-aware and semantically aligned color adaptation, and finally decoded using residual blocks, pixel-shuffle upsampling, and skip connections from the content branch to ensure structural fidelity.

By integrating spatial-channel attention into a context-aware fusion framework and introducing a consistent GT generation strategy, CANet significantly outperforms existing methods. Extensive experiments validate its superiority in visual quality and quantitative metrics, demonstrating the effectiveness of our approach.

Overall, our contributions can be summarized in three key aspects:

- We design a simple yet effective strategy for generating GT images tailored to color style transfer, addressing the core challenge of GT data scarcity and greatly simplifying the supervised training process.

- We propose the CANet network, which is the first to introduce spatial-channel attention mechanisms in the field of image color style transfer to fuse texture and color information.

- Extensive experiments demonstrate that our method consistently outperforms existing image color transfer approaches and offers a strong foundation for future research in this field.

## 2 RELATIVE WORK

### 2.1 COLOR STYLE TRANSFER

Image color style transfer aims to adapt the color characteristics of a content image to match a reference style image, while preserving structural and semantic integrity. Unlike traditional style transfer methods Chen et al. (2021); Chung et al. (2024); Zheng et al. (2024); Li (2024) which modify both style and content, color style transfer specifically targets color appearance, with an emphasis on preserving the original content.

One of the main challenges is achieving accurate color transformation while maintaining content fidelity. Recent deep learning-based approaches An et al. (2020); Chiu & Gurari (2022); Fang et al. (2024); Ho & Zhou (2021); Ke et al. (2023); Larchenko et al. (2025); Li et al. (2025); Yoo et al. (2019) have shown promising results in this area. Many convolution-based methods An et al. (2020);

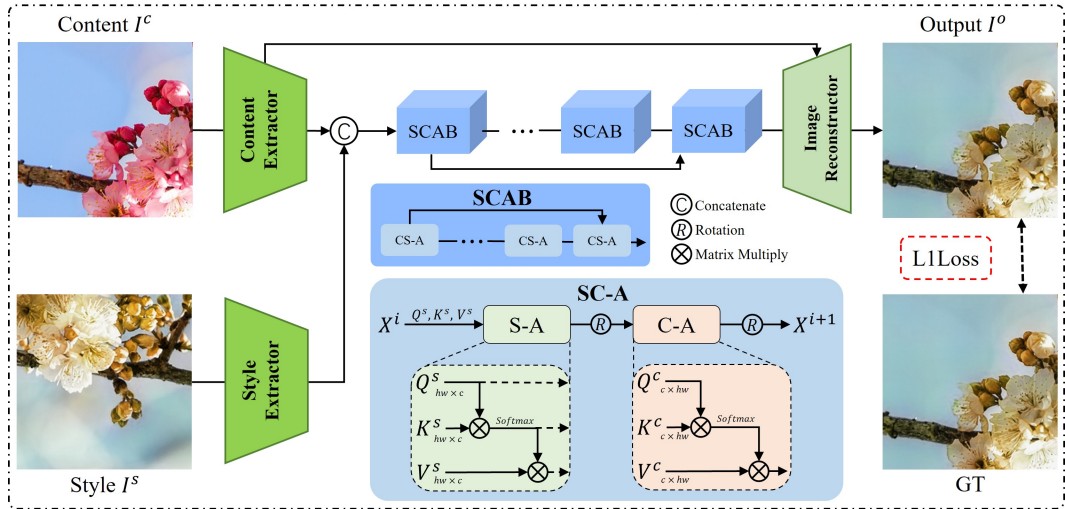

Figure 2: CANet consists of a content extraction branch and a style extraction branch, followed by four Spatial-Channel Attention Blocks (SCAB) for adaptive feature fusion, and a reconstruction module that preserves structural details while generating the final stylized output.

Chiu & Gurari (2022); Fang et al. (2024); Ho & Zhou (2021); Yoo et al. (2019)prioritize color fidelity, but often compromise structural consistency. To address this issue Ke et al. (2023) proposed a content extraction network to better preserve content. More recent work leverages optical flow Larchenko et al. (2025) or LUTs Li et al. (2025) to balance content coherence and visual realism. However, most existing methods focus on global color alignment and often overlook region-specific consistency (e.g., buildings, skies), which can result in perceptually incoherent or unrealistic outputs.

Another limitation lies in the training data. Generating well-aligned color style pairs is labor-intensive, leading to a scarcity of ground truth data, which hinders the supervised learning process and affects transfer accuracy.

## 2.2 ATTENTION MECHANISM

The attention mechanism Vaswani et al. (2017) was originally proposed in the field of natural language processing (NLP) to model long-range dependencies and capture contextual relevance between tokens. Owing to its outstanding representational capacity and ability to selectively focus on critical information, attention has become a widely adopted component in many modern deep learning architectures. With the introduction of the Vision Transformer (ViT) Dosovitskiy et al. (2020), attention-based mechanisms have been successfully extended to the computer vision domain, and have achieved performance on par with or exceeding traditional CNNs in many vision tasks.

Following this breakthrough, attention-based models have proliferated rapidly across diverse vision tasks such as image super-resolution Luo et al. (2024); Wang et al. (2024); Zhang et al. (2024), image generation Xie et al. (2025); Tian et al. (2024); Chang et al. (2022), and image enhancement Cai et al. (2023); Wang et al. (2023c); Zhang et al. (2021). These models exploit the attention mechanism's capability to dynamically reweight feature representations across spatial and channel dimensions, leading to significant improvements in visual quality and contextual consistency.

To our knowledge, spatial-channel attention has not been previously explored in the context of color style transfer. In this work, we incorporate it into the transfer framework to enable more adaptive, region-aware color fusion, resulting in improved color consistency and structure preservation.

## 3 METHODS

Recent studies, constrained by the lack of reliable GT training pairs, typically focus on precise color space mappings for color transfer, while still relying on convolution-based backbones. In this work,

we introduce a simple yet effective GT pair generation strategy. Building upon this, to address the color deviations and artifacts caused by insufficient feature fusion under the limited receptive field of convolution, we propose CANet, representing the first attempt to incorporate both spatial and channel attention mechanisms into image color transfer.

## 3.1 NETWORK ARCHITECTURE

As illustrated in Figure 2, the proposed CANet is a novel framework that consists of four key components: a content extractor, a style extractor, a spatial-channel attention block (SCAB), and an image reconstructor module.

Given a content image $I^c$ and a style image $I^s$, both in $\mathbb{R}^{C \times 16H \times 16W}$, the two images are independently processed through the content and style extractor to obtain feature maps $F^c$ and $F^s \in \mathbb{R}^{c \times H \times W}$, where $c = 64$. Both extractor share an identical architecture, each consisting of four residual blocks. At each stage, the input is downsampled by a convolutional layer with a stride of 2, reducing its spatial size by half. Although the spatial resolution is reduced, increasing the channel dimension to 64 enables the network to preserve essential content and color information.

The extracted features $F^c$ and $F^s$ are then concatenated along the channel dimension and passed into the SCAB module for deep fusion. The SCAB consists of a series of spatial-channel attention (SC-A) units with residual connections to enhance information flow and stability. The fused features are subsequently fed into the image reconstruction module, which includes four residual convolutional blocks and a final pixel shuffle layer for upsampling. Skip connections from the content features are incorporated into each stage to better retain structural information and reduce texture loss.

The overall pipeline of CANet can be formulated as:

$$F^c = f_c(I^c), \quad F^s = f_s(I^s), \tag{1}$$

$$X^1 = \text{Cat}(F^c, F^s), \tag{2}$$

$$X^{i+1} = \text{SCAB}^i(X^i), \quad i = 1, 2, \ldots, n-1, \tag{3}$$

$$I^o = f_r(X^n, F^c). \tag{4}$$

where $f_c$ and $f_s$ denote the content and style extraction branches, respectively, and $f_r$ represents the reconstruction module.

## 3.2 SPATIAL-CHANNEL ATTENTION

As illustrated in the SC-A module in Figure 2, let the input feature be denoted as $\mathbf{X}^i \in \mathbb{R}^{c \times H \times W}$, where $H$ and $W$ represent the height and width of the feature map, and $c$ is the number of channels. Subsequently, we apply a windowing mechanism to obtain $n$ feature windows $\mathbf{x}_j^i \in \mathbb{R}^{c \times h \times w}, j = 1, 2 \ldots n$, where $n = \frac{H}{h} \times \frac{W}{w}$. The query, key, and value matrices $\mathbf{Q}^s$, $\mathbf{K}^s$, and $\mathbf{V}^s \in \mathbb{R}^{hw \times c}$ are obtained via linear transformations of $\mathbf{X}^i$. The spatial attention map is then computed as $\mathbf{M}^s = \mathbf{Q}^s(\mathbf{K}^s)^{\mathrm{T}} \in \mathbb{R}^{hw \times hw}$ and used to generate the spatially attended output.

In the following stage, the matrices $\mathbf{Q}^s$, $\mathbf{K}^s$, and the attended output $(V^s)'$ are rotated to obtain $\mathbf{Q}^c$, $\mathbf{K}^c$, and $\mathbf{V}^c \in \mathbb{R}^{c \times hw}$, enabling channel-wise attention modeling. The channel attention map is then calculated as $\mathbf{M}^c = \mathbf{Q}^c(\mathbf{K}^c)^{\mathrm{T}} \in \mathbb{R}^{c \times c}$, and the channel-attended output is derived via $\mathbf{M}^c \mathbf{V}^c$. Finally, the output feature $\mathbf{X}^{i+1}$ is obtained by rotating $(V^c)'$ back to the original spatial configuration.

The overall computation process of the SC-A module can be summarized as follows:

$$\{\mathbf{x}\}_j^i = Split(\mathbf{X}^i), \ j = 1, 2 \ldots n, \tag{5}$$

$$Q^s = \mathbf{x}_j^i \times W^q, K^s = \mathbf{x}_j^i \times W^k, V^s = \mathbf{x}_j^i \times W^v, \tag{6}$$

$$M^s = Q^s \times (K^s)^{\mathrm{T}}, (V^s)' = \text{Softmax}(M^s) \times V^s, \tag{7}$$

$$Q^c = R(Q^s), K^c = R(K^s), V^c = R((V^s)'), \tag{8}$$

$$M^c = Q^c \times (K^c)^{\mathrm{T}}, (V^c)' = \text{Softmax}(M^c) \times V^c, \tag{9}$$

$$\mathbf{x}_j^i = R((V^c)'), \tag{10}$$

$$\mathbf{X}^{i+1} = Reshape(\{\mathbf{x}\}_j^i), \; j = 1, 2...n. \tag{11}$$

here, $W^q$, $W^k$, and $W^v$ represent the linear transformation matrices for the query, key, and value respectively. The operator $R(\cdot)$ denotes a rotational transpose operation. $Split$ refers to partitioning the feature map into windowed blocks, while $Reshape$ denotes restoring them to the original feature dimensions.

### 3.2.1 ACQUISITION OF TRAINING DATA

One of the major challenges in previous color style transfer methods lies in the absence of GT training pairs. In such cases, training typically relies on content and style losses for supervision, or adopts a two-stage strategy based on color space mapping.

### 3.3 TRAINING STRATEGY

When training our CANet, we prefer content and style images to share certain semantic similarities—such as grass, sky, or water—so that the model can effectively learn color transfer across corresponding regions. At the same time, training pairs with completely unrelated semantics are also necessary to ensure the model's generalization ability.

In photographic post-processing, most adjustments typically involve changes to the overall tone of the image, such as brightness, contrast, and saturation. Moreover, different regions within the same image often contain semantically related content. It is therefore reasonable to assume that the color style across different regions of a single image is generally consistent. Based on this observation, we propose a simple strategy for constructing ground-truth (GT) images for training, which naturally satisfies the above conditions, as illustrated in Figure 3. Examples of semantically related and unrelated training pairs can be found in the appendix.

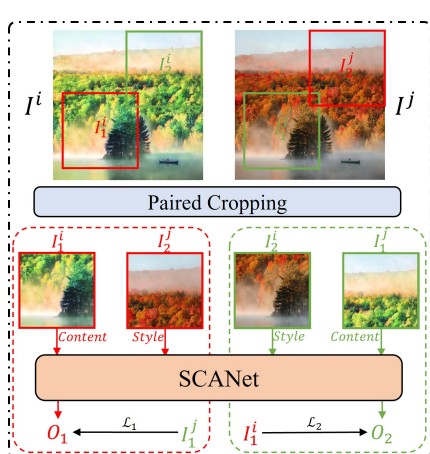

Figure 3: The training data acquisition pipeline and loss function illustration of CANet.

Specifically, $I^i$ and $I^j$ are generated from the same original image using different 3D LUTs and random color shifts. Then, $I_1^i$, $I_2^i$, $I_1^j$, and $I_2^j$ are obtained by performing two random crops from the same spatial region of $I^i$ and $I^j$, respectively.

Using these cropped images, we can form four valid training triplets consisting of a content image, a style image, and a corresponding GT image. In practice, for each original image, we select two training triplets during training: $(I_1^i, I_2^j, I_1^j)$ and $(I_1^j, I_2^i, I_1^i)$. where the first two elements are used as content and style inputs, and the third serves as the ground truth supervision image.

### 3.3.1 LOSS FUNCTION

Since GT images are available in our training setup, the design of the loss function can be significantly simplified. As illustrated in Figure 3, we adopt an L1 pixel-wise loss to supervise the output. The training objective $\mathcal{L}_t$ of our CANet is defined as:

$$\mathcal{L}_1 = ||O_1 - I_1^j||_1 \; , \; \mathcal{L}_2 = ||O_2 - I_1^i||_1, \tag{12}$$

$$\mathcal{L}_t = \mathcal{L}_1 + \mathcal{L}_2. \tag{13}$$

where $O_1$ and $O_2$ are the outputs of CANet, while $I_1^j$ and $I_1^i$ are the corresponding GT images obtained; as illustrated in Figure 3, two training pairs can be generated from each input image.

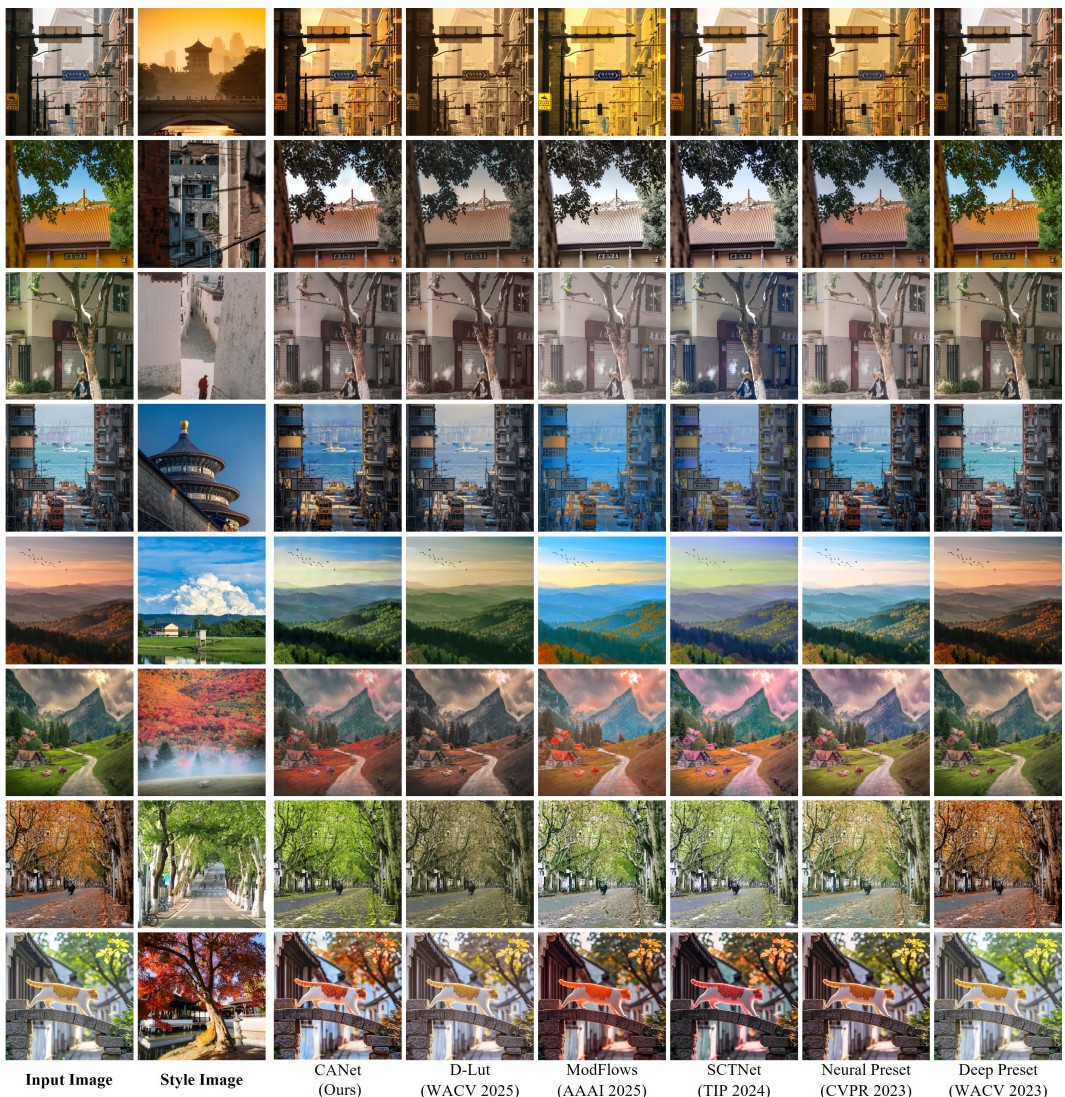

| Input Image | Style Image | CANet (Ours) | D-Lut (WACV 2025) | ModFlows (AAAI 2025) | SCTNet (TIP 2024) | Neural Preset (CVPR 2023) | Deep Preset (WACV 2023) |

Figure 4: Qualitative comparison on natural and architectural scenes. Competing methods show color deviations or artifacts, while our CANet produces more natural and semantically consistent results aligned with the reference.

## 4 EXPERIMENT

### 4.1 EXPERIMENTAL SETUP

#### 4.1.1 TRAINING DATA

We construct our training dataset using DIV2K, Flickr2K Timofte et al. (2017), and LIU2K Liu et al. (2020). During training, each image is processed to generate two different color styles. This is achieved by randomly selecting from 15 predefined 3D LUT files for initial color adjustment, followed by additional augmentations involving random modifications of hue, brightness, saturation, contrast, and color temperature to further enrich the diversity of color styles.

#### 4.1.2 TESTING DATA

Since no standardized benchmark exists for image color transfer, we construct our own evaluation set focusing on two key photography domains: natural landscapes and architectural scenes. For each

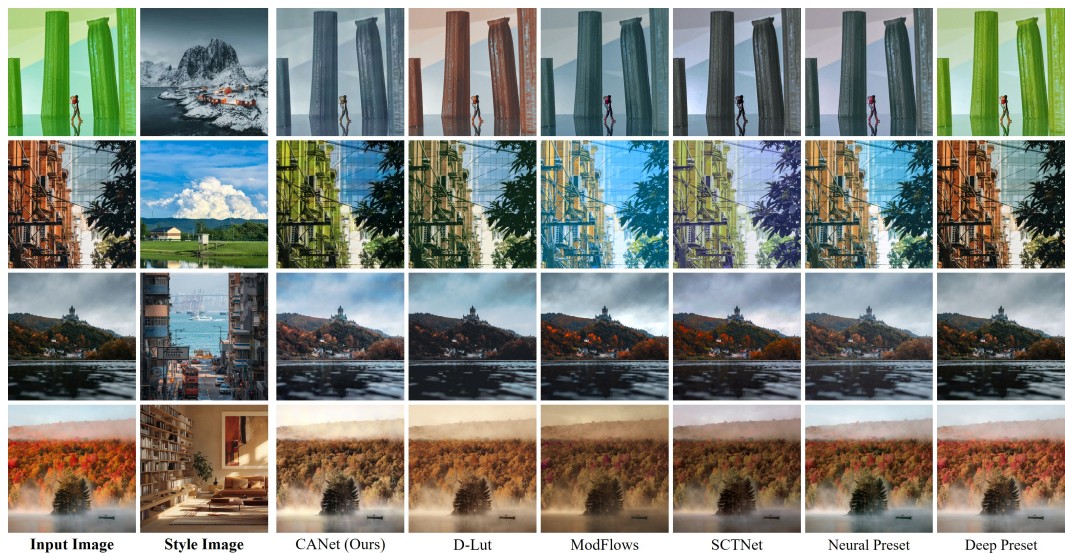

Figure 5: Qualitative comparison on the Mix test set.

domain, we select 21 images exhibiting diverse color styles, yielding a total of 420 test pairs. To further assess the generalization capability of our model, we additionally collect 68 images covering a wide range of content—such as indoor scenes, landscapes, architecture, and close-up shots—to form 1,000 test pairs, referred to as the Mix test set. All images are resized to a resolution of $1024 \times 1024$.

### 4.1.3 TRAINING DETAILS

In the design of CANet, the number of SCAB modules is set to 4, with each SCAB containing 5 SC-A blocks. The attention window size is set to 8. Regarding training parameters, we train the network for a total of 1200 epochs with an initial learning rate of 0.0001, which is halved every 400 epochs. The batch size is set to 8, and the input image size is $(512, 512)$. The optimizer used is AdamW. All training is conducted on an NVIDIA RTX 4090 GPU.

### 4.1.4 QUANTITATIVE METRICS

To objectively evaluate the performance of different methods, we adopt the Style Similarity Metric and Content Similarity Metric proposed in Ke et al. (2023), and compute their L2 distance to the ideal point $(1, 1)$, which provides a measure of how well each method balances texture preservation and style accuracy. In addition, to assess perceptual quality, we employ PI Blau et al. (2018), NIQE, BRISQUE Mittal et al. (2012), LIQE Zhang et al. (2023), and CLIPIQA Wang et al. (2023b) as no-reference image quality metrics.

### 4.1.5 COMPARISON METHODS

To validate the effectiveness of our method, we compare it against five state-of-the-art approaches employing different strategies: convolution-based Deep Preset (WACV 2021) Ho & Zhou (2021) and SCTNet (TIP 2024) Fang et al. (2024), self-supervised Neural Preset (CVPR 2023) Ke et al. (2023), flow-based Modflows (AAAI 2025) Larchenko et al. (2025), and 3D LUT-based D-Lut (WACV 2025) Li et al. (2025).

### 4.2 QUALITATIVE RESULTS

Figure 4 presents qualitative comparisons on natural landscape and architectural scenes, while Figure 5 shows results on the mixed-domain test set. As observed, Deep Preset introduces noticeable color distortions; Neural Preset improves fidelity but still suffers from incorrect transfers in sky and

Table 1: Quantitative comparison of results. The best and second-best performances are highlighted in bold and underline, respectively. Runtime is reported as the average inference time on $1920 \times 1080$ images using an NVIDIA RTX 4090 GPU. SCTNet-GT denotes the results obtained by training the SCTNet backbone with GT supervision.

| Test Set | Natural | | | | | | Architectural | | | | | | Time (s) |
|---|---|---|---|---|---|---|---|---|---|---|---|---|---|
| Metrics | Style Similarity↑ | Content Similarity↑ | Distance to (1,1) (L2) ↓ | PI↓ | NIQE↓ | BRISQUE↓ | Style Similarity↑ | Content Similarity↑ | Distance to (1,1) (L2) ↓ | PI↓ | NIQE↓ | BRISQUE↓ | |
| DeepPreset | 0.501 | **0.911** | 0.506 | 2.956 | 3.148 | **13.08** | 0.415 | **0.930** | 0.589 | 3.293 | 3.144 | **23.38** | 0.218 |
| SCTNet | 0.806 | 0.667 | 0.385 | 3.021 | 3.151 | 22.60 | 0.745 | 0.751 | 0.356 | 3.296 | 3.174 | 25.48 | 2.016 |
| D-Lut | 0.614 | 0.737 | 0.467 | 3.086 | 3.326 | 22.48 | 0.590 | 0.773 | 0.468 | 3.407 | 3.311 | 28.67 | 29.88 |
| Modflows | **0.830** | 0.670 | 0.371 | 3.030 | **3.133** | 19.28 | **0.862** | 0.705 | 0.325 | 3.288 | 3.158 | 25.83 | 1.676 |
| SCTNet - GT | 0.781 | 0.743 | 0.337 | **2.937** | 3.161 | 17.23 | 0.761 | 0.770 | 0.331 | 3.233 | 3.053 | 23.85 | 0.124 |
| **CANet (Ours)** | 0.794 | 0.772 | **0.307** | **2.934** | 3.155 | 16.67 | 0.763 | 0.789 | **0.317** | **3.225** | **3.039** | 23.56 | **0.119** |

vegetation regions. SCTNet often produces over-smoothed textures and tonal shifts, and ModFlows exhibits similar blurring and unstable adaptation. D-Lut achieves relatively stable transfer but loses translucency and fine details. In contrast, our CANet consistently generates more natural results, accurately transferring sky, grass, and foliage while preserving structural integrity. Across both semantically guided domains (landscape and architecture) and the more challenging mixed-content scenarios, CANet demonstrates superior semantic alignment and more accurate overall color adaptation.

## 4.3 QUANTITATIVE RESULTS

Table 2: Quantitative results on the Mix test set. The best and second-best performances are highlighted in bold and underline, respectively.

| Method | Style Similarity↑ | Content Similarity↑ | Distance to Oracle (L2) ↓ | LIQE↑ | CLIPIQA↑ |
|---|---|---|---|---|---|
| DeepPreset | 0.456 | **0.926** | 0.549 | **3.148** | **0.539** |
| SCTNet | 0.783 | 0.712 | 0.360 | 2.578 | 0.493 |
| D-Lut | 0.597 | 0.770 | 0.464 | 2.876 | 0.506 |
| Modflows | 0.854 | 0.713 | 0.322 | 2.730 | 0.501 |
| **CANet (Ours)** | **0.856** | 0.773 | **0.268** | 2.735 | 0.524 |

Table 1 reports quantitative comparisons on the Natural and Architectural test sets. Among all metrics, L2 distance is most critical as it reflects the balance between style similarity and content preservation: lower values indicate faithful adaptation with structural fidelity. Our CANet achieves the lowest L2 scores (0.307 / 0.317), clearly outperforming all baselines. While Deep Preset and D-Lut show high content similarity but large L2 values, and SCTNet and ModFlows emphasize style at the expense of structure, CANet delivers the most balanced results.

Moreover, our method shows clear advantages in no-reference perceptual metrics. It achieves the lowest PI scores (2.934 / 3.225), indicating outputs closer to natural images, and also surpasses others on NIQE and BRISQUE, reflecting improved naturalness and detail. Deep Preset alters images minimally, explaining its unusually high BRISQUE scores.

Table 2 presents the quantitative comparison on the Mix test set. Under the challenging mixed scenarios, our CANet achieves significantly lower L2 distances than all other methods, demonstrating its strong ability to balance style accuracy and content preservation. For LIQE and CLIPIQA, Deep Preset attains the highest scores because its outputs barely change from the inputs, leading to artificially inflated perceptual metrics. Aside from this special case, CANet ranks in the top tier across all perceptual quality metrics.

Finally, our method is highly efficient, requiring only 0.119s per $1920 \times 1080$ image. Overall, these results show that CANet achieves the best trade-off between style–content balance, perceptual quality, and efficiency, making it both effective and practical.

### 4.3.1 USER STUDY

As shown in Table 3, we randomly selected 20 groups of images from the Mix dataset, anonymized the results generated by the methods, and invited 23 volunteers to rank the six approaches. It can be observed that our method aligns well with the majority of participants' aesthetic preferences for color transfer.

| Test Set | **Ours** | Modflows | D-Lut | SCTNet | Neural Preset | Deep Preset |
|---|---|---|---|---|---|---|
| Mix | **1.39** | 1.86 | 5.00 | 2.78 | 4.30 | 5.65 |

Table 3: Average ranking of different methods in the user studies on the two test sets. A lower score indicates better subjective preference.

Table 4: Quantitative ablation study of channel attention and spatial attention. C-A denotes channel attention, and S-A denotes spatial attention. The best results are highlighted in **bold**.

| Test Set | **Natural** | | | **Architectural** | | |
|---|---|---|---|---|---|---|
| Metrics | Style Similarity↑ | Content Similarity↑ | Distance to Oracle (L2) ↓ | Style Similarity↑ | Content Similarity↑ | Distance to Oracle (L2) ↓ |
| w/o S-A | 0.666 | 0.771 | 0.404 | 0.655 | 0.784 | 0.407 |
| w/o C-A | 0.728 | 0.770 | 0.356 | 0.718 | 0.780 | 0.357 |
| **CANet** | **0.794** | **0.772** | **0.307** | **0.763** | **0.789** | **0.317** |

## 4.4 ABLATION STUDY

### 4.4.1 ABLATION STUDY ON SPATIAL-CHANNEL ATTENTION

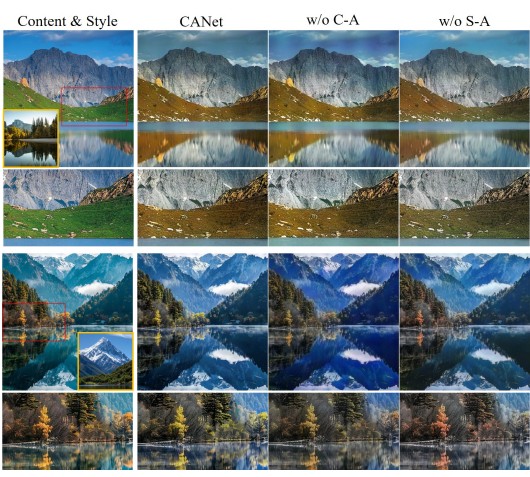

Figure 6: Qualitative Comparison of Spatial-Channel Attention. The image outlined in yellow is the target color style reference.

Image spatial attention was the earliest attention mechanism proposed in the field. It enhances feature representation by emphasizing important spatial regions. Later, researchers introduced channel attention to complement it by modeling interdependencies among feature channels, thereby improving feature fusion. Our ablation experiments (see Figure 6 and Table 4) compare these mechanisms, where C-A denotes channel attention and S-A denotes spatial attention. Results show that without channel attention, images generally preserve accurate colors but still suffer subtle deviations, and simple regions such as the sky often exhibit stripe-like artifacts. Conversely, when spatial attention is removed, images display more significant color deviations, though artifacts do not appear in simple regions like the sky. These findings highlight the complementary roles of spatial and channel attention. By integrating both, our method leverages their strengths and achieves the most consistent and visually pleasing results in image color style transfer.

### 4.4.2 IMPACT OF DIFFERENT CROPPING RATIOS IN TRAINING IMAGES

Table 5: Quantitative results on the natural dataset of models trained with different downsampled resolutions. Size denotes the target resolution after downsampling. The best performance is highlighted in **bold**, and the second-best is underlined.

| Size | Style Similarity↑ | Content Similarity↑ | Distance to (1, 1) (L2) ↓ |
|---|---|---|---|
| (1536, 2048) | 0.697 | **0.760** | 0.386 |
| (1024, 1365) | 0.703 | 0.747 | 0.390 |
| (512, 682) | 0.726 | 0.720 | 0.391 |
| (384, 512) | 0.800 | 0.736 | **0.331** |
| (256, 341) | **0.826** | 0.693 | 0.352 |

When generating training image pairs, the cropping ratio between the cropped region and the original image often affects training performance. If the cropped region is too small, the content of the content and style images may be completely unrelated; if it is too large, the content in both images may become overly similar. The former may lead to inaccurate color transfer results, while the latter may cause the model to overfit to color information and reduce its ability to generalize to diverse style images. Therefore, it is crucial to explore the optimal cropping ratio that yields the best training performance.

To reduce training time, all training images are uniformly cropped to a size of (256, 256), while the original image resolution is (1536, 2048). To simulate different cropping ratios, we downsample the original images using bicubic interpolation. The network is trained for 800 epochs with a batch size of 8, starting from an initial learning rate of 0.0001, which is halved every 200 epochs.

Our experimental results are shown in Table 5. It can be observed that as the resolution of the original image decreases, the model tends to focus more on learning color information. However, when the style and content images become too similar, the model's ability to generalize to diverse style images is reduced, ultimately leading to increased content loss in the generated results. Therefore, it is essential to strike a balance between color fidelity and content preservation during training in order to achieve the best overall performance. Our experiments indicate that an optimal cropping ratio is achieved when the cropped image size is approximately half that of the original image.

### 4.4.3 Effect of GT-Supervised Training

To further validate the effectiveness of GT-supervised training and demonstrate the relative performance of our method, we emphasize that most existing image color transfer approaches do not release training code, making direct reproduction infeasible. To ensure a fair comparison, we adopt the backbone network from Fang et al. (2024) as our feature fusion module and retrain it under our GT supervision framework, using exactly the same hyperparameters as our method. As shown in Table 1, SCTNet-GT significantly outperforms the original training scheme across most quantitative metrics, confirming the effectiveness of introducing GT pairs. More importantly, our method consistently surpasses SCTNet-GT on all evaluation metrics, highlighting the strength of our network architecture in fully leveraging GT supervision.

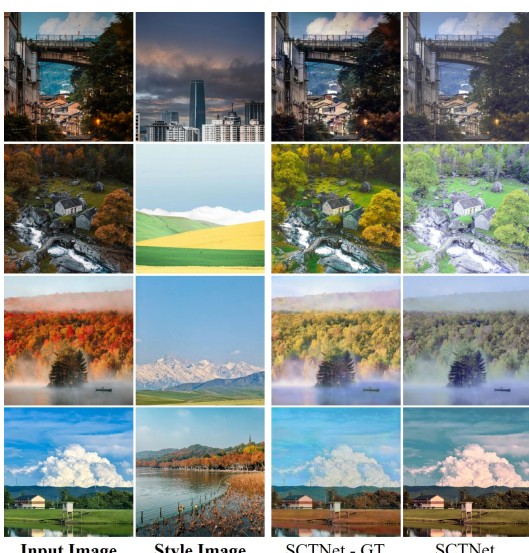

Input Image    Style Image    SCTNet - GT    SCTNet

Figure 7: Qualitative Comparison of SCTNet-GT and SCTNet. -GT denotes training with our GT pairs.

Figure 7 provides visual evidence of these findings. Compared with the original SCTNet, SCTNet trained with GT supervision produces noticeably more accurate colors and better structural alignment, reducing both global deviations and local artifacts. Nevertheless, our method further improves color fidelity and perceptual quality, producing results that are more natural and stylistically consistent with the reference. Taken together, these experiments provide strong evidence of the effectiveness of our GT-based training strategy and demonstrate the superiority of our proposed network architecture.

## 5 Conclusion

In this paper, we propose two key innovations. First, we introduce a simple yet effective strategy for generating GT images, which significantly reduces the complexity of training design and alleviates the burden of loss function tuning. Second, we incorporate spatial-channel attention mechanisms into the image color transfer task for the first time, overcoming the limitation of previous methods that focused mainly on global color consistency. By jointly modeling spatial and channel-wise dependencies, our CANet enables precise and region-specific color transfer between semantically corresponding regions in the content and style images, such as sky to sky or grass to grass, thereby enhancing the visual coherence of the results. Extensive experiments show that CANet consistently achieves superior qualitative and quantitative performance, while also delivering the fastest inference time (0.119s per 1920 × 1080 image), making it both effective and efficient for image color style transfer.

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

## A APPENDIX

### A.1 EXAMPLES OF TRAINING PAIRS WITH DIFFERENT CONTENT RELEVANCE

As shown in Figure 8, randomly cropped images enable the model to learn styles with varying degrees of semantic similarity during training, as they naturally include fully related, partially related, and completely unrelated cases.

### A.2 RESULTS WITH DIFFERENT 3D LUTS

Using multiple 3D LUTs during training helps simulate real-world color distributions. In our experiments, we initially employed 15 LUT files, and the results are shown in Figure 9.

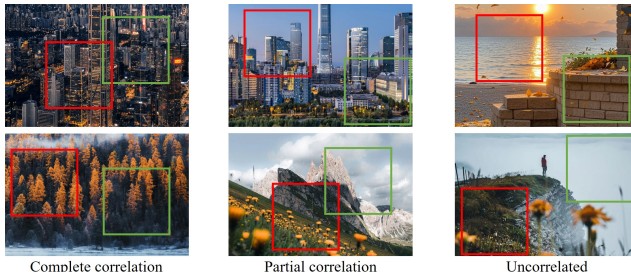

Complete correlation     Partial correlation     Uncorrelated

Figure 8: Examples where the content and style images are semantically fully related, partially related, and completely unrelated.

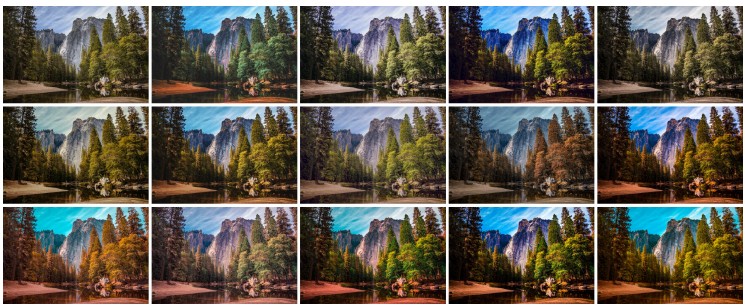

Figure 9: The figure presents results obtained using the 15 LUTs adopted in our training.

## A.3 DISCUSSION

### A.3.1 DIFFERENCE FROM OTHER GT GENERATION STRATEGIES

Before the GT strategy proposed in this work, several existing methods had also adopted ground-truth–based supervision, such as Zhao et al. (2021); Lee et al. (2020a) and Ke et al. (2023); Lee et al. (2020b). However, as illustrated on the left side of Figure 10, methods like Zhao et al. (2021); Lee et al. (2020a) obtain style images by applying random distortions and color variations to the GT image. This forces the content and style images to share the same semantics, limiting the model's generalization ability and preventing it from learning meaningful cross-scene color transfer.

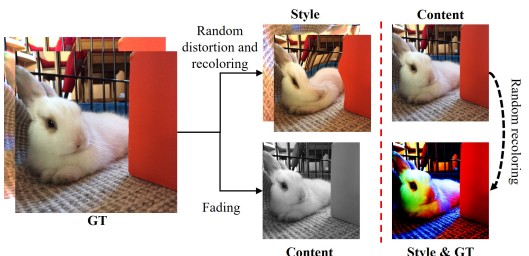

Figure 10: Different training data generation strategies.

As shown on the right side of Figure 10, methods such as Ke et al. (2023); Lee et al. (2020b) generate style images by randomly altering the color tone of the content image and then using these as GT targets. This strategy requires aggressively compressing the style information, leaving the model with only limited color cues to learn from. As a result, semantic information is lost, and the final color transfer becomes misaligned with the semantic structure of the input.

Our GT generation strategy is illustrated in Figure 3. We first enhance the images using 15 different 3D LUTs together with random adjustments of hue, brightness, and saturation. As shown in Figure 8, we then randomly crop different regions from the same source image to construct the content, style, and GT triplets. Owing to this randomness, the generated content–style pairs naturally exhibit three types of semantic relationships, which greatly improve the model's generalization ability. Moreover, this design removes the need for aggressively compressing the style image, allowing the model to retain both color and semantic information during training.

### A.3.2 COLOR TRANSFER AND STYLE TRANSFER

Although color transfer and style transfer appear similar, they differ fundamentally in one key aspect: **whether the texture information of the content image is altered**. As illustrated in Figure 11, we compare our task with the CVPR 2025 style transfer model SaMam. It is evident that SaMam significantly modifies the texture and fine details of the content, which is clearly unsuitable for color transfer in photographic images.

Although color transfer and style transfer are two distinct visual tasks, it is still meaningful to discuss how the attention mechanism used in our color transfer framework differs from the attention designs in prior style transfer works Tumanyan et al. (2022); Deng et al. (2022). Color transfer is primarily used in photographic post-processing, where the computational capability of photographers' devices can vary widely. Therefore, efficiency and inference speed are central considerations in our attention design.

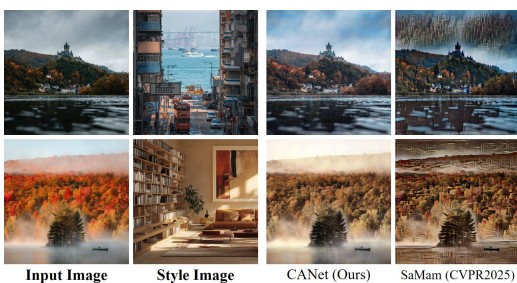

**Input Image**    **Style Image**    CANet (Ours)    SaMam (CVPR2025)

Figure 11: Qualitative comparison with the style transfer method SaMam.

As shown in the Figure 12, in contrast to ViT-based attention modules commonly used in style transfer Tumanyan et al. (2022); Deng et al. (2022), our pixel-wise attention is significantly more computationally efficient. Since color transfer does not require altering texture information—and typically performs much lighter modifications to the content image compared with style transfer—we argue that adopting a ViT-style attention architecture is unnecessary and even unsuitable for this task.

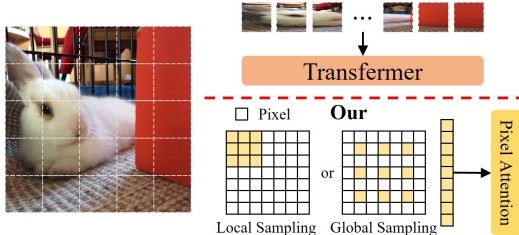

Figure 12: Comparison between our block-wise attention and pixel-wise attention.

Although attention mechanisms have been widely explored in style transfer, color transfer and style transfer are distinct tasks. To the best of our knowledge, no prior work has applied attention specifically to color transfer. Our method is therefore the first to introduce an attention-based formulation for this task.

**Complexity of block attention vs. point-wise attention.** To make the computational difference explicit, we compare a ViT-style *block attention* with a *point-wise attention inside blocks* under the same setting. Let the input feature be $X \in \mathbb{R}^{64 \times 64 \times 64}$, and we partition it into non-overlapping $8 \times 8$ blocks. The number of blocks is

$$B = \frac{64}{8} \cdot \frac{64}{8} = 64,$$

and each block contains $S = 8 \times 8 = 64$ spatial locations.

**ViT-style block attention.** For block attention, each $8 \times 8$ block is flattened into a single token of dimension

$$D_{\text{block}} = 8 \times 8 \times 64 = 4096.$$

Treating these $B = 64$ tokens as a sequence, ViT-style block attention performs global self-attention among all blocks. The dominant FLOPs come from $QK^\top$ and the multiplication with $V$, giving

$$\text{FLOPs}_{\text{block-attn}} = 2 \cdot B \cdot D_{\text{block}}^2 = 2 \cdot 64 \cdot 4096^2.$$

Numerically,

$$\text{FLOPs}_{\text{block-attn}} = 2.15 \times 10^9.$$

**Point-wise attention inside blocks.** For point-wise attention, self-attention is applied independently inside each $8 \times 8$ block. Each block contains $S = 64$ pixel tokens, each with dimension

$C = 64$. The cost per block is

$$\text{FLOPs}_{\text{per-block}} = 2 \cdot S \cdot C^2 = 2 \cdot 64 \cdot 64^2.$$

Aggregating over all $B = 64$ blocks yields

$$\text{FLOPs}_{\text{point-attn}} = B \cdot 2 \cdot S \cdot C^2 = 2 \cdot 64^4 = 3.36 \times 10^7.$$

**Comparison.** Thus, under the same block partition (64 blocks of size $8 \times 8$), ViT-style block attention is approximately

$$\frac{2.15 \times 10^9}{3.36 \times 10^7} \approx 64\times$$

more expensive than point-wise attention inside blocks. The difference arises because block attention builds global interactions among 4096-dimensional block tokens, while point-wise attention only models the 64 pixel tokens within each local block.

### A.4  STYLE AND CONTENT SIMILARITY METRIC

#### A.4.1  STYLE SIMILARITY METRIC

Following the metric design of Ke et al. Ke et al. (2023), style similarity is evaluated using a learned discriminator that measures the consistency of color style between two images. The discriminator is trained on a large collection of human–retouched color–style categories, and outputs a score in the range $[0, 1]$, where higher values indicate stronger color–style agreement. Unlike simple color–statistic comparisons, this learned metric captures higher–level style attributes such as global contrast and overall tonal consistency, making it more reliable for assessing color–style alignment.

#### A.4.2  CONTENT SIMILARITY METRIC

For content preservation, we follow the recommendation of Ke et al. Ke et al. (2023) and compare edge maps extracted from the two images. Instead of the commonly used HED detector, a more accurate edge extraction method (e.g., LDC) is employed to obtain fine structural details. The content similarity score is then computed as the SSIM between the two edge maps, providing a structure–focused measurement of semantic and geometric consistency.

