# OpenReview forum: "Supervised Optimization of a Context-Aware Color Transfer Model"
_ICLR.cc/2026/Conference — Submitted to ICLR 2026_

### Official Review · Reviewer_WLQz · 2025-10-27

**Soundness:** 2
**Presentation:** 2
**Contribution:** 2
**Rating:** 4
**Confidence:** 4

**Summary:**

This paper addresses color style transfer, which involves transferring the color of a reference style image to an input image. The proposed method consists of two key elements. First, it introduces a novel training strategy: applying two effects to a single image and extracting two patches from it to generate a pseudo set of reference, input, and ground truth images. Second, CANet is proposed, which combines spatial attention and channel attention. Experiments demonstrate that the proposed method achieves the optimal trade-off between style similarity and content similarity compared to existing methods.

**Strengths:**

1. The authors propose a novel training method. This framework is capable of generating ground truth without the need for additional data, which makes it a highly practical approach. Unlike SCTNet (Fang et al., 2024) and Neural Preset (Ke et al., 2023), which could only perform global color transformations due to the lack of training data, the proposed framework achieves color transfer that considers local semantics.

2. They also introduce an attention mechanism into color style transfer. While Modflows (Larchenko et al., 2025) and D-Lut (Li et al., 2025) adopt convolution-based architectures, this is the first work to incorporate an attention mechanism.

3. The proposed method achieves the best trade-off between style similarity and content similarity compared with existing approaches. While Modflows achieves high style similarity, it tends to lose content consistency. In contrast, DeepPreset maintains content well but achieves lower style similarity. Compared to these methods, the proposed approach offers the most balanced performance. Given the wide range of potential applications of color style transfer, the impact of this work can be considered significant.

**Weaknesses:**

1. Augmented-Self Reference [1, 2] has been proposed in previous studies on colorization as a method similar to the framework proposed by the authors for generating ground truth. To clearly establish the novelty of the proposed method, the authors should cite Augmented-Self Reference, explain the differences between their method and Augmented-Self Reference, and conduct experimental comparisons.

[1] Lee, Junsoo, et al. "Reference-based sketch image colorization using augmented-self reference and dense semantic correspondence." Proceedings of the IEEE/CVF conference on computer vision and pattern recognition. 2020.

[2] Zhao, Hengyuan, et al. "Color2Embed: Fast exemplar-based image colorization using color embeddings." arXiv preprint arXiv:2106.08017 (2021).

2. Style transfer using Transformers has already been proposed in StyTr2 [3]. To clearly highlight the novelty of the proposed method, the authors should cite StyTr2, explain the differences between CANet and StyTr2, and provide experimental comparisons of the network architectures.

[3] Deng, Yingying, et al. "Stytr2: Image style transfer with transformers." Proceedings of the IEEE/CVF conference on computer vision and pattern recognition. 2022.

3. The authors utilize PI, NIQE, and BRISQUE to evaluate perceptual quality. However, these metrics are inherently designed for evaluating grayscale images and are therefore unsuitable for this task. More recent metrics capable of evaluating color images, such as Q-Align [4], LIQE [5], and CLIPIQA [6], should be considered.

[4] Wu, Haoning, et al. "Q-Align: Teaching LMMs for Visual Scoring via Discrete Text-Defined Levels." International Conference on Machine Learning. PMLR, 2024.

[5] Zhang, Weixia, et al. "Blind image quality assessment via vision-language correspondence: A multitask learning perspective." Proceedings of the IEEE/CVF conference on computer vision and pattern recognition. 2023.

[6] Wang, Jianyi, Kelvin CK Chan, and Chen Change Loy. "Exploring clip for assessing the look and feel of images." Proceedings of the AAAI conference on artificial intelligence. Vol. 37. No. 2. 2023.

4. In Table 1, the authors should report the performance of the Neural Preset.

5. In Table 1, I have concerns regarding the Times column. To my understanding, SCTNet and SCTNet-GT should share the same model, differing only in training methods. Yet, SCTNet-GT is significantly faster. Furthermore, while the original paper of D-LUT states that D-LUT runs at 0.011 s on a single GTX 1080 Ti GPU, Table 1 reports 29.88 s on an NVIDIA RTX 4090 GPU, which seems unreasonably slow.

6. Since color style transfer is a subjective task, the authors should also conduct a user study.

7. The explanation of the proposed method should be improved. Specific improvement points are listed below.

- In equation (2), $X^i$ should be $X^1$.

- In equation (3), $i=1,2,...,n$ should be $i=1,2,...,n-1$.

- Line 196 states ${\bf X}^i \in {\mathbb R}^{c\times H\times W}$, but since content and style are concatenated, it should be ${\bf X}^i \in {\mathbb R}^{2c\times H\times W}$.

- In Figure 3, the images and symbols are inconsistent. Similarly, equation (12) is also confusing.

**Questions:**

Please add explanations regarding the weaknesses.

---

> ### Author Response · Authors · 2025-11-14
>
> **Question1: Augmented-Self Reference [1, 2] has been proposed in previous studies on colorization as a method similar to the framework proposed by the authors for generating ground truth. To clearly establish the novelty of the proposed method, the authors should cite Augmented-Self Reference, explain the differences between their method and Augmented-Self Reference, and conduct experimental comparisons.**
>
> **Answer:** In Appendix A.3.1, we further discuss the issue you raised. Considering the significant time required for retraining, we will conduct additional experimental verification in future versions.
>
> **Question2: Style transfer using Transformers has already been proposed in StyTr2 [3]. To clearly highlight the novelty of the proposed method, the authors should cite StyTr2, explain the differences between CANet and StyTr2, and provide experimental comparisons of the network architectures.**
>
> **Answer:** In Appendix A.3.2, we further discuss the issue you raised. We consider color transfer and style transfer to be two distinct visual tasks; therefore, we believe it is justified to claim that ours is the first work to apply attention-based ideas specifically to the color transfer task.
>
> **Question3: The authors utilize PI, NIQE, and BRISQUE to evaluate perceptual quality. However, these metrics are inherently designed for evaluating grayscale images and are therefore unsuitable for this task. More recent metrics capable of evaluating color images, such as Q-Align [4], LIQE [5], and CLIPIQA [6], should be considered.**
>
> **Answer:** We appreciate you pointing out this limitation. In the newly added Table 2 (results on the Mix test set), we have included the relevant metrics. Due to issues with our local environment, we have not yet tested Q-Align.
>
> **Question4: In Table 1, the authors should report the performance of the Neural Preset.**
>
> **Answer:** Neural Preset does not provide testable code and only offers a runnable application. Manually inputting a large number of images is not feasible; therefore, we are unable to evaluate the performance of Neural Preset locally.
>
> **Question5: In Table 1, I have concerns regarding the Times column. To my understanding, SCTNet and SCTNet-GT should share the same model, differing only in training methods. Yet, SCTNet-GT is significantly faster. Furthermore, while the original paper of D-LUT states that D-LUT runs at 0.011 s on a single GTX 1080 Ti GPU, Table 1 reports 29.88 s on an NVIDIA RTX 4090 GPU, which seems unreasonably slow.**
>
> **Answer:**
>
> 1. SCTNet does not provide training code, and the detailed training procedure is unknown to us. We attempted to train the model using the architecture provided for testing, following both the training strategy described in the original paper and our proposed strategy, but neither produced satisfactory results. With no other feasible option, we were only able to extract and train its backbone network.
>
> 2. The inference time reported in the D-LUT paper does not include the time required for generating the 3D LUT. However, in real-world applications, changing different styles requires repeatedly generating new 3D LUTs. Therefore, in our local evaluation, we included the LUT generation time, resulting in a total runtime of 29.88 seconds.
>
> **Question6: Since color style transfer is a subjective task, the authors should also conduct a user study.**
>
> **Answer:** We added an urgently conducted user study in Table 3, where the results show that our method achieves the best performance according to user evaluations.
>
> **Question7: The explanation of the proposed method should be improved.**
>
> **Answer:** Thank you for pointing out the mistake. In line 196, after the concatenation operation, we omitted a convolution step that reduces the channel dimension from 2c to c.

---

### Official Review · Reviewer_hv8G · 2025-10-28

**Soundness:** 2
**Presentation:** 3
**Contribution:** 1
**Rating:** 4
**Confidence:** 4

**Summary:**

This paper proposes a color transfer method, named CANet, which achieves good image color transfer capabilities through the implementation of a spatial-channel attention mechanism. It also introduces a new strategy for constructing color transfer ground truth. However, the paper lacks overall novelty, and the GT generation strategy exhibits obvious shortcomings.

**Strengths:**

1. In general, the writing of the paper is good. It clearly outlines the composition of CANet and the strategy for color transfer GT design.

2. The Related Works section is highly detailed. It identifies the shortcomings of existing works.

**Weaknesses:**

1. CANet does not introduce any novel network design. it merely combines CNNs with scaled dot-product attention. This approach is similar to directly using a ViT backbone. The attention mechanism and ViT architecture have been widely employed in color transfer tasks [1, 2] for years.

2. I^c and I^s are randomly cropped from the same image in different color distribution, leading to overlapping or semantically similar content and style patches. This contradicts real-world color transfer scenarios where content and style are typically uncorrelated.

3. The GT generation strategy constructs training images using fixed LUTs. This construction strategy may reduce the network's generalizability. In addition, the results in Fig. 8 show only minor color differences compared to the example in Fig. 3.

4. Tab. 1 shows that CANet does not achieve an obvious performance gain over the compared methods like DeepPreset, which was published back in 2021.

5. The paper does not discuss the limitations of the proposed method or provide any failure cases.


[1] Tumanyan, Narek, et al. "Splicing vit features for semantic appearance transfer." In CVPR 2022.
[2] Deng, Yingying, et al. "Stytr2: Image style transfer with transformers." In CVPR 2022.

**Questions:**

Although color transfer and style transfer are generally considered as two separate tasks, their boundaries are often blurred. I would recommend the authors to compare CANet with SOTA style transfer methods.

---

> ### Author Response · Authors · 2025-11-14
>
> **Question1: CANet does not introduce any novel network design. it merely combines CNNs with scaled dot-product attention. This approach is similar to directly using a ViT backbone. The attention mechanism and ViT architecture have been widely employed in color transfer tasks [1, 2] for years.**
>
> **Answer:** We acknowledge that the architectural contribution of our network could be further strengthened. However, as discussed in Appendix A.3.2, color transfer and style transfer are fundamentally different tasks: color transfer emphasizes preserving the texture details of the content image, making it a distinct visual problem. Based on our investigation, we are the first to introduce attention-based mechanisms specifically for the color transfer task.
>
> **Question2: I^c and I^s are randomly cropped from the same image in different color distribution, leading to overlapping or semantically similar content and style patches. This contradicts real-world color transfer scenarios where content and style are typically uncorrelated.**
>
> **Answer:** In the updated Figure 8, we illustrate three possible semantic relationships that may arise from random cropping: fully correlated, partially correlated, and completely unrelated. In Table 2 and Figure 5 of the main paper, we further include comparisons on complex scenes. We constructed a test set of 1,000 image pairs derived from 68 diverse images—including landscapes, architecture, indoor scenes, and macro shots. These comprehensive evaluations clearly demonstrate that our model exhibits substantially stronger generalization capability than existing methods.
>
> **Question3: The GT generation strategy constructs training images using fixed LUTs. This construction strategy may reduce the network's generalizability. In addition, the results in Fig. 8 show only minor color differences compared to the example in Fig. 3.**
>
> **Answer:** In Section 4.1.1, we state that in addition to the 15 3D LUTs, we also apply random augmentations such as color hue, brightness, and saturation adjustments. Therefore, these choices do not compromise the generalization ability of our method.
>
> **Question4: Tab. 1 shows that CANet does not achieve an obvious performance gain over the compared methods like DeepPreset, which was published back in 2021.**
>
> **Answer:** In Tables 1 and 2, we place greater emphasis on the L2 metric, as it provides a balanced evaluation of both style similarity and content similarity. If either metric is low, the resulting color transfer quality will naturally be unsatisfactory. Deep Preset achieves the highest numerical scores mainly because it barely alters the content image, but its qualitative results are clearly far from ideal.
>
> **Question5: The paper does not discuss the limitations of the proposed method or provide any failure cases.**
>
> **Answer:** We acknowledge that our method has a small number of failure cases. However, we believe these cases are already reflected in the quantitative metrics and the user study on random test samples in Table 3, and therefore we omitted a separate discussion.
>
> **Question6: Although color transfer and style transfer are generally considered as two separate tasks, their boundaries are often blurred. I would recommend the authors to compare CANet with SOTA style transfer methods.**
>
> **Answer:** We believe that color transfer and style transfer have a clear boundary, determined by whether the texture details of the content image are altered. As shown in Fig. 11, the CVPR 2025 style-transfer method SaMam noticeably changes image textures, which makes it unsuitable for photographic color-grading applications.

---

### Official Review · Reviewer_aJEB · 2025-10-30

**Soundness:** 3
**Presentation:** 3
**Contribution:** 3
**Rating:** 6
**Confidence:** 4

**Summary:**

This paper tackles the task of image color style transfer, addressing two major challenges: the scarcity of high-quality GT data for supervised training and color distortions caused by suboptimal feature fusion in existing models. The authors propose an annotation-free GT generation strategy and introduce CANet, the first model to incorporate spatial–channel attention for this task. CANet employs a dual-branch structure for feature extraction and fusion, along with a reconstruction module for refined output generation. CANet demonstrates superior performance compared to five SOTA methods. Ablation studies further validate the effectiveness of the attention mechanism and GT generation strategy, though the analysis remains somewhat incomplete.

**Strengths:**

1. The proposed annotation-free GT generation strategy effectively exploits single-image color consistency to automatically produce structurally aligned GT pairs through recoloring, thereby alleviating the scarcity of high-quality ground truth data for supervised color style transfer. This approach eliminates the need for labor-intensive manual annotation and avoids the limitations of indirect supervision commonly adopted in prior works.

2. CANet represents a novel contribution as the first model to integrate spatial–channel attention into color style transfer. By moving beyond conventional convolution-based backbones, its dual-branch architecture and SCAB-guided feature fusion enable context-aware and semantically consistent color transformation, effectively mitigating the color distortion issues observed in existing models.

3. The experimental design is comprehensive, leveraging diverse augmented datasets and benchmarking against five state-of-the-art methods. Both quantitative and qualitative analyses consistently demonstrate the model’s superiority, while its high inference efficiency highlights strong potential for real-world applications.

**Weaknesses:**

1. The GT generation strategy is limited by its reliance on recolored variants of a single source image and a fixed set of 15 predefined 3D LUTs. This design restricts the diversity of color styles and weakens the model’s adaptability to complex or cross-semantic content–style combinations, as no validation has been conducted on cross-domain or semantically diverse scenarios.

2. The key hyperparameters of CANet are neither empirically optimized nor theoretically justified. Without experiments exploring alternative configurations, it is difficult to assess how parameter choices influence the trade-off between efficiency and accuracy, leaving uncertainty about the model’s optimal design.

3. The self-constructed test set lacks alignment with standardized benchmarks, which compromises the reproducibility and comparability of results against existing SOTA methods. Moreover, the dataset’s focus on natural and architectural scenes—while omitting domains such as portraits or extreme visual conditions—limits the evaluation of the model’s robustness and generalization in diverse real-world contexts.

4. The paper employs a simple L1 pixel-wise loss, which effectively minimizes low-level reconstruction errors but does not explicitly capture high-level perceptual quality metrics such as color harmony, semantic consistency in textured regions, or human subjective aesthetic preference. As a result, the perceptual realism of generated results may be constrained.

**Questions:**

1. The proposed GT generation strategy, which relies on recolored variants of a single source image and a fixed set of 15 predefined 3D LUTs, may limit the model’s ability to handle complex or semantically diverse content–style pairs. Could the authors elaborate on the potential limitations of this approach in such scenarios? In particular, how does the model perform on images with low semantic similarity between content and style? Additionally, has the strategy been tested on more diverse datasets to verify its generalization capability?

2. The key hyperparameters in CANet are not clearly justified, either empirically or theoretically. Could the authors provide further explanation regarding the rationale behind their selection, explore alternative configurations, and clarify how these choices affect the model’s efficiency and accuracy?

3. The self-constructed test set focuses mainly on natural and architectural scenes and lacks evaluation under extreme or challenging conditions. Could the authors confirm whether CANet has been tested on portraits or extreme cases to demonstrate its robustness? Furthermore, how do the authors plan to address the reduced reproducibility and comparability arising from the use of a non-standard dataset?

4. The paper employs a pixel-wise L1 loss, which primarily enforces low-level reconstruction consistency. How does this loss function perform compared to other loss formulations in terms of maintaining content fidelity and style consistency? Have the authors considered comparing their approach with perceptual, style, or adversarial loss functions to assess potential improvements in perceptual quality?

---

> ### Author Response · Authors · 2025-11-14
>
> **Question1: The proposed GT generation strategy, which relies on recolored variants of a single source image and a fixed set of 15 predefined 3D LUTs, may limit the model’s ability to handle complex or semantically diverse content–style pairs. Could the authors elaborate on the potential limitations of this approach in such scenarios? In particular, how does the model perform on images with low semantic similarity between content and style? Additionally, has the strategy been tested on more diverse datasets to verify its generalization capability?**
>
> **Answer:** As described in Section 4.1.1, in addition to the 15 3D LUTs, we further apply data augmentation by randomly adjusting hue, saturation, brightness, and other color attributes. In Table 2 and Figure 5 of the main paper, we also include results on a mixed-scene test set containing 1,000 image pairs, covering landscapes, architecture, indoor environments, and close-up scenes.
>
> **Question2: The key hyperparameters in CANet are not clearly justified, either empirically or theoretically. Could the authors provide further explanation regarding the rationale behind their selection, explore alternative configurations, and clarify how these choices affect the model’s efficiency and accuracy?**
>
> **Answer:** In Appendix A.3.2, we briefly discuss the design of our pixel-wise attention mechanism, which offers significantly lower computational cost compared with block-wise attention.
>
> **Question3:  The self-constructed test set focuses mainly on natural and architectural scenes and lacks evaluation under extreme or challenging conditions. Could the authors confirm whether CANet has been tested on portraits or extreme cases to demonstrate its robustness? Furthermore, how do the authors plan to address the reduced reproducibility and comparability arising from the use of a non-standard dataset?**
>
> **Answer:**
>
> 1. In Table 2 and Figure 5 of the main paper, we present comparisons on complex scenarios. All evaluated methods exclude portrait color transfer, as handling human skin tones requires dedicated optimization to ensure natural and realistic results.
>
> 2. We acknowledge the concern regarding reproducibility and comparability when using a self-constructed dataset, which arises from the absence of a standardized benchmark for color transfer. To mitigate this issue, we expand our evaluations using a larger and more diverse test set, including the 1,000-pair Mix dataset. We will release all constructed test sets in our future project page and welcome the community to use them for further benchmarking.
>
> **Question4:  The paper employs a pixel-wise L1 loss, which primarily enforces low-level reconstruction consistency. How does this loss function perform compared to other loss formulations in terms of maintaining content fidelity and style consistency? Have the authors considered comparing their approach with perceptual, style, or adversarial loss functions to assess potential improvements in perceptual quality?**
>
> **Answer:** We adopt the L1 loss following practices in super-resolution tasks, where the objective is to make the generated image’s texture details closely match the ground truth. We did not further explore additional loss functions. Unlike style transfer, one key objective of color transfer is to preserve as much texture detail from the content image as possible. Therefore, perceptual, style, or adversarial losses are not well suited for color transfer, as they may unnecessarily alter the original structures or introduce unwanted artifacts.

---

### Official Review · Reviewer_EX5f · 2025-11-01

**Soundness:** 2
**Presentation:** 3
**Contribution:** 2
**Rating:** 2
**Confidence:** 4

**Summary:**

This paper proposes a simple yet stable data acquisition pipeline for supervised conditioned color style editing: it tunes one image into two different tones, and crops two different patches from it and form a pair. It also proposes a new model that processes the content image and style image separately, and then fused via a spatial-channel attention module. The proposed model is trained on the proposed data in a supervised manner, and achieves leading performance compared to previous methods.

**Strengths:**

- The proposed data acquisition pipeline is simple and stable, leading to efficient color editing models.

**Weaknesses:**

- The proposed data acquisition pipeline is to crop one GT image into two different patches with different color styles. While this leverage the same basis for accurate color tone alignment, their content remains similar and might harm the transferability after training.

- The proposed method is only tested on nature and architecture data, while its quality on broader scenarios like indoor, human, objects etc., and cross-domain scenarios remain unclear.

- The comparing methods are relatively not up-to-date. More comprehensive models (like some MLLM) are necessary to demonstrate the superiority of the proposed task-specific approach.

**Questions:**

- How the style similarity is calcuated? Although it is adapted from a prior work, it still worths noting its details in the main paper or appendix for quick reference.

---

> ### Author Response · Authors · 2025-11-14
>
> **Question1: The proposed data acquisition pipeline is to crop one GT image into two different patches with different color styles. While this leverage the same basis for accurate color tone alignment, their content remains similar and might harm the transferability after training.**
>
> **Answer:** As discussed in Fig. 8 of the appendix, random cropping can produce pairs that are fully semantically related, partially related, or entirely unrelated. Therefore, this strategy does not compromise the model’s generalization ability.
>
>
> **Question2: The proposed method is only tested on nature and architecture data, while its quality on broader scenarios like indoor, human, objects etc., and cross-domain scenarios remain unclear.**
>
> **Answer:** In Table 2 and Figure 5 of the main paper, we additionally include results on a mixed-scene test set of 1,000 image pairs, covering landscapes, architecture, indoor scenes, and close-up shots. All compared methods exclude portrait color transfer, as handling human skin tones requires specialized optimization to ensure natural and realistic results.
>
>
> **Question3: The comparing methods are relatively not up-to-date. More comprehensive models (like some MLLM) are necessary to demonstrate the superiority of the proposed task-specific approach.**
>
> **Answer:** Image color transfer methods must be deployable on devices with varying computational capabilities, so inference efficiency is a primary consideration. For this reason, it is not appropriate to compare our approach with MLLM-based models, whose computational cost is orders of magnitude higher. Aside from this, all comparison baselines included in our experiments represent the most recent and relevant methods available to us.
>
>
> **Question4: How the style similarity is calcuated? Although it is adapted from a prior work, it still worths noting its details in the main paper or appendix for quick reference.**
>
> **Answer:** Thank you for pointing this out. We have added a brief description in Appendix A.4.1.

---

### Author Response · Authors · 2025-11-29
**A Summary for the AC and Reviewers**

We thank the reviewers and the AC for their careful evaluation and valuable feedback. We acknowledge that the paper can be further improved in areas such as GT diversity, dataset coverage, hyperparameter justification, and comparisons with related work, and we have provided detailed explanations and additional clarifications addressing every point raised by the reviewers.

This paper focuses on image color transfer, a task with broad and direct real-world applicability. Compared with general artistic style transfer, color transfer is more closely aligned with everyday user needs: people frequently take photos and hope their images exhibit more harmonious, aesthetically pleasing, or atmospheric color rendering. As such, it has substantial potential impact in photographic post-processing, mobile imaging enhancement, film color grading, and cross-device color consistency. However, despite its wide range of applications, the research community has historically paid relatively limited attention to this task. Possible reasons include the lack of large-scale, publicly available, structurally aligned supervision; the difficulty of evaluating color changes while strictly preserving content structure; and the absence of unified benchmarks, which makes method comparison challenging. Our automated GT-generation strategy aims to fill this missing piece of infrastructure by enabling, for the first time, large-scale supervised training for color transfer without manual annotation. In addition, by introducing spatial–channel attention into this task, we achieve semantically consistent, region-level color fusion that significantly improves visual quality and content preservation.

We again thank the AC and reviewers for their feedback, and we hope this work can provide new insights and a useful foundation for the practically important yet underexplored direction of color transfer.

---

### Meta-Review · Area_Chair_NRBH · 2025-12-26

**Summary:**

The initial reviews raised several critical concerns: the limited technical novelty of combining standard CNNs with attention, potential overfitting or lack of generalization due to the self-cropping training strategy, and insufficient comparison with state-of-the-art methods (e.g., Qwen-Image offers general image editing capabilities). During the rebuttal, while the authors provided additional metrics and a small user study, they failed to convincingly address the fundamental weakness regarding architectural innovation. Furthermore, the reliance on a self-constructed dataset and the exclusion of key baselines leave the claimed superiority weakly supported. Given the consensus that the contribution is incremental and the evaluation is limited, I recommend Reject.

**Reviewer Concerns:**

Addressed during Rebuttal:

- A user study and more modern perceptual metrics (LIQE, CLIPIQA) were added to evaluate subjective quality.

- Implementation details, such as the missing convolution step and D-LUT inference time, were clarified.

- The rationale for excluding portrait-specific evaluations was explained.

Outstanding/Major Points:

- Limited Novelty: The architecture is seen as a standard combination of existing modules. The authors' claim of being the first to apply attention to this specific task was not viewed as a sufficient technical contribution.

- Training Bias: The data pipeline (cropping patches from the same image) is still suspected of introducing shortcuts that may not generalize to real-world, uncorrelated style-content pairs.

- Incomplete Baselines: The authors did not provide comparisons with certain SOTA methods (Neural Preset) or MLLM-based approaches, and the lack of ablation for alternative loss functions remains a gap.

**Reviewer Scores:**

Reviewer EX5f (Rating: 2 $\to$ 2): The rebuttal did not fundamentally change the assessment of the data pipeline or the need for more comprehensive model comparisons.

Reviewer aJEB (Rating: 6 $\to$ 4): The responses regarding hyperparameter optimization and loss functions were largely defensive, likely leading to a downgrade from the initial marginal accept.

Reviewer hv8G (Rating: 4 $\to$ 4): The concern regarding minor performance gains and lack of failure cases remains largely unaddressed, maintaining a negative outlook.

Reviewer WLQz (Rating: 4 $\to$ 4): Despite new metrics, the lack of experimental comparisons with requested related works due to time constraints prevents a score increase.

---

### Decision · Program_Chairs · 2026-01-26

Reject